The potential for income improvement and biodiversity conservation via specialty coffee in Ethiopia

Schuit Pascale 1
http://orcid.org/0000-0002-5513-3615 Moat Justin 2
http://orcid.org/0000-0002-7621-2873 Gole Tadesse Woldemariam 3
Challa Zeleke Kebebew 4
Torz Jeremy 1
Macatonia Steven 1
Cruz Graciano 5
Davis Aaron P. 6 a.davis@kew.org
1 Union Hand-Roasted Coffee , London , UK
2 Biodiversity Informatics and Spatial Analysis, Royal Botanic Gardens, Kew , Richmond, Surrey , UK
3 Environment and Coffee Forest Forum (ECFF) , Addis Ababa , Ethiopia
4 UNOCHA , Addis Ababa , Ethiopia
5 HiU Coffee , Panama City , Panama
6 Natural Capital & Plant Health, Royal Botanic Gardens, Kew , Richmond, Surrey , UK
Ghermandi Andrea
Electronic publication date: 2021 Feb 9
Publication date: 2021
Volume: 9
Electronic Location ID: e10621
Received 2020 Jul 31; Accepted 2020 Nov 30
Copyright: © 2021 Schuit et al.
Copyright year: 2021
Copyright holder: Schuit et al.
License: This is an open access article distributed under the terms of the Creative Commons Attribution License, which permits unrestricted use, distribution, reproduction and adaptation in any medium and for any purpose provided that it is properly attributed. For attribution, the original author(s), title, publication source (PeerJ) and either DOI or URL of the article must be cited.
License URL: https://creativecommons.org/licenses/by/4.0/

Keywords: Biodiversity, Livelihoods, Coffee, Ethiopia, Forest, Remote sensing, Deforestation, Specialty coffee

Funding: Darwin Initiative (UK; DFiD) 2 2-006 Amar-Franses and Foster-Jenkins Trust This work was supported by the Darwin Initiative (UK; DFiD), ref. no: 2 2-006, and the Amar-Franses and Foster-Jenkins Trust. The funders had no role in study design, data collection and analysis, decision to publish, or preparation of the manuscript.

==============================
The specialty coffee sector represents opportunities for coffee farmers and other actors due to increased value within the supply chain, driven by elevated consumer purchase prices. We investigated these relationships up to the export stage, as well as the potential for specialty coffee to improve stakeholder participation in biodiversity conservation. Household data was collected from a sample of 272 coffee farmers belonging to five primary cooperatives, in the heavily forested area of Yayu, in the Illubabor administrative zone of Oromia regional state in south-western Ethiopia, for the 2017/2018 harvest season. Qualitative and quantitative data, gathered between 2015 and 2019, from focus group discussions, was used to supplement the survey and to explain the empirical findings. We show that the income from coffee (and thus household income) can be increased, in some cases substantially, via participation in the specialty coffee market. The unit price for coffee was much higher than standard market prices and those including certification premiums. Whilst quality is a key factor for specialty coffee, income increases via the specialty market are caveat bound. In particular, there is a critical requirement for efficient and effective cooperative management, and we provide suggestions for improvements. In the long-term, more focus is needed to increase the financial and human capacities of Ethiopian coffee cooperatives, to make them more effective partners in the specialty coffee value chain. The high conversion rates from harvested to clean exportable specialty coffee (8:1, or higher) is identified as a major constraint on profitability. We show that specialty coffee can bring about positive environmental benefits. Elevating coffee prices and farm profitability to bring household income levels to around or above the global poverty line may ensure the continuation of forest coffee production, and thus the retention of forest. The increases in income via specialty coffee production, reported in the study, were achieved without increasing land use or the application of environmentally costly inputs. Moreover, analysis of satellite data shows that a large proportion of the coffee farms surveyed retain a level of forest cover and quality approaching primary (undisturbed) forest, and that the coffee production area at Yayu has not experienced any significant deforestation (since 2000). We recommend that coffee premiums linked to environmental benefit should demonstrate clearly defined and appropriate metrics, as we have demonstrated here for forest (canopy) quality and coverage (area).

Introduction

Environmentalists desire to expand protected areas and reduce the intensity of farming; agriculturalists attempt to increase crop production in order to enhance food security and income for farming communities (Scherr & McNeely, 2003). This conflict of interest is often most acute when there is a desire to resolve poverty alleviation and preserve remaining biodiversity (Sanderson & Redford, 2003). These two interests have rarely been well integrated (Scherr & McNeely, 2008). Preservation of natural habitat via protected area designation brings key benefits for biodiversity conservation, and the sustainability of ecosystem services (e.g., preservation of watersheds, soil functioning, climate amelioration and pollinator services), but can, for example, reduce available resources, such as space for farmland expansion, which may aggravate poverty. This leads to a vicious circle, since impoverished stakeholders are less likely to invest in environmental conservation (Holden, Shiferaw & Wik, 1988) even if they are aware of, and require, natural landscapes as an integral part of their agroecological environment. In order to address the incongruity between agriculture and biodiversity conservation, it has been argued that they should be combined using ecoagricultural approaches (Scherr & McNeely, 2008), where environmental conservation and livelihoods improvement and sustainability are mutually compatible.

The coffee sector represents an interesting case in this discussion, because coffee production is both a major cause of biodiversity loss (Gobbi, 2000), mainly through deforestation (Laakkonen, 1996) and, conversely, an agent of biodiversity (and forest) preservation (Gove et al., 2008; Hylander et al., 2013). Given the amount of land dedicated to coffee production in the tropical regions of the world, which would be almost exclusively forested in their natural state, the impact of coffee farming over the last 150 years alone has been substantial. However, the global impact of coffee farming on deforestation is not well publicized, and scientific studies measuring its impact are scarce. At the same time it is recognized that ecoagricultural (or agroecosystem) coffee production can provide useful biodiversity habitats (Matson et al., 1997).

In Ethiopia, forest canopy (shade coffee) is required for successful cropping, and is the main system used for coffee production (Davis et al., 2018; Gole, 2003; Gole et al., 2008; Moat et al., 2017; Schmitt, 2006; Schmitt et al., 2009). Ethiopia holds a special position in the world of coffee, as Arabica is indigenous only to the highlands of Ethiopia and a small area of neighboring South Sudan (Davis et al., 2012). Most of the coffee production systems in Ethiopia are based on modified natural forest, and a large proportion of those are home to wild Arabica coffee. The semi-domestication of local and regionally specific Arabica populations for coffee farming has resulted in a wide and complex range of flavor profiles, providing Ethiopian coffee with its unique selling point (Davis et al., 2018). Also of considerable note is that forested areas in Ethiopia without coffee production have a higher deforestation rate and risk than those with coffee production, although the biodiversity value of forest production areas is lower than natural, undisturbed forest (Hylander et al., 2013; Schmitt et al., 2009). In this way, coffee production can serve as a good example of ecoagriculture, that is, generating co-benefits for production, and thus local stakeholders, and biodiversity (Scherr & McNeely, 2008). Recent downturns in coffee prices, and, overall, cyclic market volatility (International Coffee Organization, 2016, 2020), threaten these ecoagricultural systems due to low or negative profitability and market sustainability. When producers (farmers) are forced to transition away from ecoagricultural approaches (e.g., shade or forest coffee) due to lack of profitability, they are often faced with alternatives that do not require forest cover, resulting in a loss of biodiversity, ecosystem services and biomass. This example exemplifies the close association between poverty and biodiversity conservation.

In the late 1990s and early 2000s, coffee certification schemes were seen as part of the solution for smallholder poverty, and since then certification has been widely adopted in the coffee sector. Whilst coffee certification has brought beneficial financial, social (including governance, health and safety) outcomes for many farmers (Darko, Lynch & Smith, 2017; Giovannucci & Potts, 2008; Mitiku et al., 2017; Van Rijsbergen et al., 2016), some contest that the benefits may be ambiguous (Bacon, 2005), variable across farms and communities (Giovannucci & Potts, 2008), with uneven economic advantages (Haight, 2011), of meager benefit (Minten et al., 2018), of negligible financial or social impact for cooperatives (Jena et al., 2012; Ortiz-Miranda & Moragues-Faus, 2015), and make no positive difference (relative to other forms of employment in the production of the same crops) for wage workers (Cramer et al., 2014). There have also been concerns regarding coffee quality due to the mechanisms involved in certification (Haight, 2011; Tolessa, Duchateau & Boeckx, 2018). Very few studies have investigated the influence of certification on biodiversity improvement. Of those studies that are available, only minimal positive differences in biodiversity and soil health have been identified (Bacon, 2005; Cramer et al., 2014; Giovannucci & Potts, 2008). Certification schemes linked directly to promoting biodiversity appear more likely to bring about positive benefits for biodiversity; those covering biodiversity or environmentally friendly coffee (Gobbi, 2000; Gove et al., 2008), shade coffee certification (Mas & Dietsch, 2004; Takahashi & Todo, 2017) and bird friendly coffee (Mas & Dietsch, 2004) report positive benefits for biodiversity.

A number of coffee companies employ their own forms of certification, or bypass it altogether by adopting a direct trading approach. These programs operate with different names, including Direct Trade, Responsible Sourcing, and Ethically Sourced. As with certification, these initiatives aim to contribute towards poverty alleviation. For these privately developed schemes, however, there is a concern that they may not offer benefits to producers, or may only poorly mimic effective sustainability practices. Claims of income improvements are often stated, but they may be ambiguous and unsubstantiated. Peer reviewed research on Direct Trade mechanisms for coffee are scarce. In one study, using data from the US and Scandinavia (MacGregor, Ramasar & Nicholas, 2017), it was found that the benefits were skewed towards the consumer segment of the value chain (e.g., roasters and retailers), optimized for firms’ private interest for taste quality and improved profitability, rather than producer benefit and the public interest in sustainable development.

Irrespective of certification, improved coffee value chains can represent opportunities for local producers (and value chain representatives) in developing countries, through access to higher value markets, new technologies and sustainable market access (Lutz & Olthaar, 2017; Lutz & Tadesse, 2017). In the case of the specialty coffee value chain, there can be further advantages, as higher market prices can be achieved due to the retail price differential brought about by the increase in quality and improved customer experience. In the ideal model, the consumer-paid price differential brought about by higher quality is passed back to the producers, although improvements in producer incomes are often ambiguous and unsubstantiated, and some even argue that there are negative consequences (Fischer, 2017). This situation may be exasperated in those cases where producers do not: have the ability to assess the quality (and thus the monetary value) of their coffee via sensory and quality evaluation; are unaware of the structure, processes and actors involved in the value chain; or are plainly deceived at some point along the value chain.

Specialty coffee is high quality coffee, with a higher purchase value compared to other coffee. Specialty coffee has seen continued growth since 2000 (Adroit Market Research, 2019; Bacon, 2005; Ponte, 2002) and is anecdotally reported to comprise c. 10% of the global coffee market, although the actual proportion of production (export) and value of specialty coffee sector in the global market is unclear and difficult to measure. By comparison, 55% of global coffee production (as reported in 2016/17) is certified in some fashion, of which 20% is procured as standard-compliant coffee by the industry (Panhuysen & Pierrot, 2018).

Given that the unit price (US$/lb) of specialty coffee may far exceed those of commodity coffee, by up to two or three fold, and in rare cases many times more, how does specialty coffee influence farmer income? In contrast to the large number of studies measuring the outcomes of certification for coffee producers (see above), there has been very little research on the impact of specialty coffee on farm income and profitability. Indeed, there has been a critical lack of peer-reviewed papers on coffee production costs and profitability across the entire coffee sector (Montagnon, 2017). In addition, since biodiversity preservation may be directly linked to income, as discussed above, what impact might increases in income brought about by the specialty coffee value chain have on biodiversity?

In this contribution we investigate the coffee value chain for a specialty coffee produced in Ethiopia, using a case study, to address three specific questions: (1) Can coffee income (and thus household income) for small-holder farmers be improved via progression to the specialty coffee market? (2) Are there specific factors governing the unit price ($/lb) paid to farmers and their profitability? (3) How might improvements in income and profitability from specialty coffee align with forest (and thus biodiversity) conservation? To our knowledge there is no research detailing the income from coffee earned by small-scale coffee producers in Ethiopia, and scant information about how the value chain is linked to land use. For our household coffee income calculations we used a household data survey of 272 coffee farmers in the Illubabor zone of south-western Ethiopia, collected in 2017 and 2018, in combination with various qualitative and quantitative data gathered between 2015 and 2019 from various sources (including primary and secondary cooperative data, and purchase and exportation documents). For assessments of forest (canopy) quality and coverage we used geolocated point data (GPS coordinates), remote sensing data and historical forest-cover data (2001–2018).

Data collection and methodology

Survey area

The project site was located in south-western Ethiopia in the Oromia region, Illubabor administrative zone of Oromia regional state, within the southern part of the UNESCO registered Yayu Coffee Forest Biosphere Reserve (Fig. 1). The production method used at Yayu falls under the Forest Coffee (FC) system, or more precisely covers a range of intensifications of the Semi-forest Coffee and Forest Garden Coffee systems, where the canopy is largely retained or thinned to provide the correct level of shade, the understorey is cleared or partially so, and coffee is planted at commercially viable densities (Davis et al., 2018). The project site covers five primary cooperatives, located over seven kebeles (the smallest administrative units in Ethiopia; given in parentheses): Achebo (Achebo), Gechi (Gechi and Bondawo), Geri (Geri), Wutate (Wutate), and YayuZuria (Wabo and Hamuma). The coffee farms (340 GPS locations) were located over an elevations range of 1,379–1,841 m, with a mean elevation of 1,559 m.

Figure 1 Map of Yayu Coffee Forest Biosphere Reserve and coffee farm locations.

Showing boundaries for the three zones (core area, buffer and transition zones), main settlements (white text), major vegetation types, and coffee farm locations. A simple classification has been applied to 2017 Landsat EVI image (see inset). Note: these classes are only used for this visualization and not for the analysis (Fig. 2).

The Yayu Coffee Forest Biosphere Reserve (167,000 ha) is divided into: (1) core area, (2) buffer zone, and (3) transition zones(s), as shown in Fig. 1. It is home to around 450 higher plants, 50 mammal, 200 bird, and 20 amphibian species, plus important wild crop genetic resources for Arabica coffee (Coffea arabica) (Gole, 2003; Gole et al., 2008). Coffee cultivation occurs almost predominately within the forested areas of the buffer and transition zones, with scattered farms in the agricultural zone; coffee cultivation or other agricultural activities are not allowed in the core area (Fig. 1). At Yayu, coffee generates around 70% of the cash income for over 90% of the population, and for some farmers, coffee provides almost all household income (Gole, 2003).

The five (primary) coffee cooperatives (Achebo, Gechi, Geri, Yayu Zuria and Wutate) are only allowed to grow coffee within the boundaries of their Kebeles, and within a Kebele it is not permitted to have more than one cooperative with the same mandate. Yayu Zuria is an exception, which has members of both Hamuma and Wabo kebeles. The cooperatives are organized and controlled by their members and are of relatively small scale and managed on a voluntary basis. The households in the five kebeles employ various livelihood strategies. They do not only cultivate coffee but also maize, spices, and (the legal narcotic) khat (Catha edulis). Ethiopia’s society is characterized by complex social and political relationships. The coffee sector is directed and dependent upon different institutions: Ethiopia Coffee and Tea Authority, Cooperative Promotion Agency, and various agricultural extension offices. These agencies have different layers at national, regional, zonal and woreda level. The Ethiopian Commodity Exchange is responsible for sensory evaluation and grading of the coffee, and acts as the official trading platform for coffee.

In 1999, the Ethiopian government established Coffee Farmers’ Cooperative Unions to manage coffee export business on behalf of primary coffee cooperatives that lacked human resources and logistic capacity. The first coffee cooperative union, the Oromiya Coffee Farmers’ Cooperative Union (OCFCU), was established to unite primary cooperatives in the Oromiya Region. Coffee unions are allowed to trade outside state-run coffee auctions; private traders are obliged to participate in coffee auctions if trading is not undertaken via a union. Operating outside the auction system means that cooperatives can export directly to buyers on the world market. The five primary cooperatives in this study exported their coffee via the OCFCU until 2015; in 2016 they exported their coffee through Sorgeba Union, a local union that had been active in the area distributing fertilizer and commercializing maize for primary cooperatives. Sorgeba Union had no previous export experience in coffee or any other crop when they obtained their export license in 2016.

In general, the study area has a very low level of public services and infrastructure, including transport facilities, schools, health clinics, and reliable sources of drinking water, and there are no regular transport services around the cooperatives as most journeys are undertaken by foot. These limitations pose serious challenges for improving the livelihoods of the coffee-growing community at Yayu.

Data collection

The empirical study was conducted at several levels. Different sample units were selected from managerial and administrative levels in the coffee value chain as well as from smallholder coffee producers in different coffee cooperatives. Field visits were undertaken at least twice a year from 2014 until 2018, to document activities and developments through an in-depth approach involving qualitative and quantitative interviews with producers, cooperative leaders, government representatives and buyers. The extended nature of this research enabled assessment of the coffee value chain for the five cooperatives of study, which would have otherwise be difficult to capture. The main data collection from household surveys was carried out from January 2017 to March 2018.

Our study employed a mix of quantitative and qualitative methods. Structured interviews were conducted with a total of 272 coffee producing smallholders, each a member of one of the five selected cooperatives. The questionnaire used in the survey gathered information on three main topics, including: (1) household characteristics (e.g., income, expenditure); (2) production methods; and (3) income (sales) of coffee. The interview form, which provides the details of the subjects and questions asked, is given in the Supplemental Files. In addition, expert interviews were conducted with cooperative heads and other cooperative representatives. To supplement these data, we applied focus group discussions (FGDs) with the board of directors of the cooperatives. Semi-structured guidelines were used to facilitate the discussions. For the purposes of this study we mainly used production method (see below) and income.

All five cooperatives had obtained a Rainforest Alliance certificate, which expired in June 2016. For sampling we used the Rainforest Alliance membership list, which included both active and non-active members (e.g., delivering and not delivering coffee to the primary cooperatives). Purposive sampling techniques were employed to obtain data from members having their coffee plots near the village centers, as well as members living close to, or in, the buffer zone. The list includes the name and the size of the plot. The location of each farm was not included in the Rainforest Alliance list; geographical coordinates were added by using a GPS (see below). The primary cooperatives involved in the survey, the number of members, and respondents of the survey (number and percentage) are given in Supplemental Files.

Terms and conventions, calculation of coffee prices, coffee processing conversions

In this article we use a number of terms and conventions that are widely used in the coffee sector but not in the scientific literature, and some description is thus required. The term “cherry” refers to the fruit of the coffee, including the bean (the seed), its fleshy covering (mesocarp) including the skin (pericarp). It may be either “fresh cherry”, referring to recently harvested fruits, or “(sun)dried cherry”, the state it reaches after being dried in the sun for several days to a few weeks. Following drying the coffee is milled (mechanically) to remove the dried pulp and skin (mesocarp) and the parchment (endocarp), a hard and crispy (when dry) layer that covers the seed. In “washed coffee” the cherry is first de-pulped using specific machinery and then fermented in water tanks to remove the remaining pulp, dried (in the sun), and then milled to remove the parchment. Both processing techniques, sundried and washed, produce “green” or “clean” coffee as the final product, at least as far as production is concerned, and the state in which it is shipped, before roasting. This is also linked to the term green bean equivalent (GBE), which states the equivalent amount in the green or clean state. Further details, specific to coffee processing in Ethiopia can be found in Davis et al. (2018).

Coffee prices are given in US$ per pound ($/lb) for clean/green coffee (the standard currency for global trading in coffee), and US$ per kilogramme (US$/kg) for cherry (fresh or dried). The Ethiopian Birr (ETB) is the unit of currency for Ethiopia. Farming productivity in given in kg per hectare (kg/ha).

Throughout the process of coffee production, from cherry to clean coffee, the crop reduces drastically in weight. This is measured as a ratio and is referred to as either the conversion or conversion rate. Standard conversion rates are: for fresh cherry to sundried cherry 3:1, and for sundried cherry to clean coffee 2:1; (Davis et al., 2018; International Trade Centre, 2011); for fresh cherry to clean coffee, via the washed coffee processing method, is often stated to be around 5:1, but in Ethiopia is usually around 6:1 (Davis et al., 2018; Minten et al., 2018). So, for example, it would take 6 kg of fresh cherry to produce 1 kg of clean coffee.

The primary cooperatives (five in our case study) undertake the farming of coffee and consolidate the fresh cherry for processing and drying. The secondary cooperative (called a union in Ethiopia) undertakes the export logistics, dry milling management, sorting and grading, and export. The final price at time of export is known as the Free on Board (FOB) price, sometimes referred to as the Freight on Board price, which is the amount received by the secondary cooperative, that is, the price paid by the buyer at point of export. The buyer is responsible for the shipping and all other fees associated with logistics of receiving goods at their premises. This price is expressed as US$ per lb ($/lb) of clean exportable (green) coffee. The farm gate price is the price received by the farmers for their coffee; in Ethiopia this is generally given in ETB per kg of fresh or sundried cherry. The primary cooperatives have neither an export license nor the human/financial capital to export coffee; this service is provided by the secondary cooperative (the union).

Specialty coffee is high quality coffee, which is assessed using specific physical grading criteria and sensory (aroma and taste) evaluation, as defined by the Specialty Coffee Association (SCA) (Specialty Coffee Association, 2020). A standard international quality score is applied, where coffees with 80 or more points (out of 100) are classified as specialty coffee. In specialty coffee no primary defects and five secondary defects are allowed (Specialty Coffee Association, 2020). Defects in coffee affect the flavor of the coffee, and therefore there is a direct relationship between the number and type of defects and coffee quality, as assessed by the cupping score. Defects can occur naturally but are mainly due to external factors, such as bad management (e.g., poor calibration of the various processing machines, poor hygiene, and improper storage), unripe harvesting, flower fertilization problems, and insect damage, amongst others.

Focus data collection: production method and income

The average price for fresh cherry, sundried cherry and washed clean coffee was asked in the survey, although the vast majority of farmers did not answer this question (e.g., “Don’t remember”), and thus these data could not be collected via this method. Instead, we obtained the price of fresh cherry and washed coffee as purchased by the five primary cooperatives, via their financial records; and the average price of sundried cherry was taken from FGDs on prices. Using the survey data, we then calculated the average amount (percentage) of coffee being processed by each of three scenarios, across each of the five cooperatives: to give the amount (average kg per farm) and price of coffee (US$/kg and US$/lb) for coffee sold as: sundried cherry to either middlemen or the primary cooperative, or fresh cherry for washing (for specialty coffee). The three scenarios were: (1) Baseline scenario—non-specialty coffee (100% of coffee sundried); (2) Improved scenario—25% specialty coffee (25% washed; 75% sundried); and (3) 100% specialty coffee scenario (100% washed). For washed coffee we ascertained both the commodity price paid (ICO price based on the price for Colombian milds (International Coffee Organization, 2020)) and the specialty coffee price paid (i.e., by Union Hand-Roasted Coffee, London, UK) via a direct trade mechanism.

Data limitations

The reliability of the data is subject to the usual caveats that apply when information is based on recollection. The majority of the respondents had limited education, and keep minimal or no record of their production, or other financial metrics. Therefore, it can be assumed that the figures reported by them are not a 100% accurate. Self-reported measures of total income are unlikely to be reliable, because, quite apart from an understandable reluctance to reveal such information to a stranger (Morris et al., 2000), the myriad transactions undertaken by such self-employed people make it unlikely that respondents know these data (Deaton, 1997). Additionally, they might elicit patterns of under-reporting as well as over-reporting, for socially undesirable and desirable behaviors and attitudes, respectively. This is particularly the case in households with irregular income sources or among individuals who engage in self-employment, and intentionally overstate or understate their income. We believe that this danger of over and/or under-reporting was minimized by the familiarity of the interviewer, and the purpose of the study/project in the area. Moreover, in this study, data from the household survey was used in combination with data from the cooperative (on cherry purchase from farmers) to validate answers and for analyses.

Coffee trees are prone to a biennial bearing pattern, particularly in Ethiopia (Davis et al., 2018), where a high production year alternates with a low production year (Wintgens, 2004; Wrigley, 1988). For the coffee purchase data collection period the harvest (late 2017/early 2018) coincided with an “off” year in the biennial cycle. Variances in weather can also have a marked influence, for example, drought episodes, as well as pest and disease outbreaks. Coffee revenue is calculated by the price (US$/lb) multiplied by the quantity (kg per household), with the amount sold to the cooperative varying from year to year. Coffee income results will thus differ from one year to another, and sometimes markedly. Even though income data from a single year has its limitations, it is still immensely valuable data, and can be compared and scaled against data from additional years, particularly if the key variables are known (e.g., yield, position in the biennial bearing cycle, knowledge of coffee values over the value chain), as was the case in our study. In October 2017, Ethiopia’s central bank devalued the Ethiopian birr by 15 percent, from 23 BIRR to 27 BIRR for 1 USD, its first such move in 7 years, as a means of boosting lagging exports.

Geographical data and remote sensing

Satellite imagery

For spatial analysis of the Yayu Coffee Forest Biosphere Reserve and surrounding area we used Landsat 8 Collection 1 Tier 1 (Chander, Markham & Helder, 2009) Annual Enhanced Vegetation Index (EVI) Composite imagery in Google Earth Engine (Gorelick et al., 2017). EVI is generated from the near-IR, red and blue bands of each Landsat image, and ranges in value from −1.0 to 1.0 (Huete et al., 2002). EVI is similar to Normalised Difference Vegetation Index (NDVI) but is more robust for areas of humid vegetation where NDVI values generally becomes oversaturated. Due to the large amounts of cloud cover over our study site we used a composite image which removes cloud cover by overlaying all images (24 images) to give a yearly mean, whilst discounting cloud cover (for the 2017 project year).

Location data, park boundaries and other zonation

Farm location data for coffee farms (Fig. 1) was collected via GPS (Garmin eTrex 20x) during dedicated farm visits. We recorded the name of the farmer and the geographical coordinates of the farm. A total of 340 sites were recorded from across the five Yayu primary cooperatives of the study area, with multiple points for some of the 272 respondent sites (e.g., those with large farms). The main boundaries of the Yayu Coffee Forest Biosphere Reserve, core area, transition and buffer zones (Fig. 1) files were obtained from Environment and Coffee Forest Forum, in Ethiopia. To provide an additional area for comparison to our coffee farm sites and the main zones of the Yayu Coffee Forest Biosphere Reserve, we digitized the main agriculture zones from Google Earth® (Google Earth, 2019), and defined this as the Yayu Agricultural Zone, as shown by white boundary lines (Fig. 1).

Spatial analysis

To assess forest (canopy) quality/health we used Landsat imagery EVI, processed in Google Earth Engine (Gorelick et al., 2017) and exported to ArcMap 10.7 (Esri, Redlands, CA, USA) for visualization and additional spatial analysis. For our site locations, we extracted the EVI pixel values by means of cubic convolution using the 16 nearest cell centers (approximately 70 m around the original points). For the reserve boundaries (core area, buffer and transition zones) and the agricultural zone, previously identified from Google Earth® (Google Earth, 2019), were sampled using the nearest pixel. EVI pixel values were export to R (R Core Team, 2019) and plotted as violin plots with the ggplot package (Wickham, 2016). For visualization purposes (Fig. 1), a simple classification was applied to the 2017 Landsat EVI image, where >0.2 = grass/scrubland/urban, 0.2–0.4 = poor vegetation generally crops and agriculture, 0.4–0.6 transition forest/forest, >0.6 forest, although it should be made clear that these classes were not used in the analysis (Fig. 2).

Figure 2 Violin plot of EVI values for Yayu Coffee Forest Biosphere Reserve and coffee farm locations.

Core area, buffer, and transition zones, Yayu agriculture zone, and coffee farm location data (340 geolocated data points). See Fig. 1 for location of zones and data points. See Table 3.

To assess forest cover we queried Global Forest Watch (Global Forest Watch, 2014) for all our sample areas and points, between the years 2001 and 2018, to gather a comparison of deforestation rates and forest coverage. To test that coffee farms were not a random sample of the other zones, we applied a T-test for the mean EVI of coffee farms vs other zones in R (R Core Team, 2019).

Results

Descriptive results

The majority of the interviewed households depend on coffee as their main source of cash income (70% or higher), in agreement with previous studies (Gole, 2003). Other sources of income are grains, livestock, fruits, and the legal narcotic khat. Of the 272 respondents 239 (88%) were male and 33 (12%) were female; the average age was 48 (male 49; female 45). Only 19% of the respondents reported as having an education higher than primary school, and 19% reported as having received no education at all. A little under half (47%) of the respondents reported not having a phone (45% of males and 64% of females).

Of the 272 respondents, 15 provided incomplete answers on production and coffee sales; elimination of these respondents left 257 for calculating household productivity, although neither reducing respondent number nor using missing values resulted in negligible differences (e.g., within 1 kg of clean coffee (GBE) production per household). The average farm size was 2.35 ha, with 1.75 ha devoted to coffee (74% of the land). The smallest coffee farm was a 0.25 ha and the largest 17 ha. In the study area, farmers aim to plant approximately 3,000 coffee trees per ha, but the actual number is often lower due to the open nature of the coffee trees being grown (i.e., they are not compact, like many improved Arabica cultivars), and because of the space taken up by shade trees. The average yield for the 2017/2018 harvest season was 200 kg per ha (of clean exportable coffee (GBE)), or an average of 350 kg per farm (200 kg × 1.75 ha); this was an “off” (unproductive) year within the biennial cycle. From ongoing field trials in Yayu (Davis, 2013–2018, personal observation) and anecdotal information from of our survey, we see that in an “on” year yield can be as high as 650 kg/ha, or perhaps even 750 kg/ha, but as low as 207 kg/ha. There was considerable variation in yield across the five cooperatives, during both “on” and “off” years. Moreover, there is no guarantee that a projected “on” year will materialize. In the harvest year following 2017/18, our study period and an “off” year, there was a lack of rain for the projected “on” year harvest (2018/19) and yields were even lower than the previous (“off”) year. We estimate than an average yield for “on” year across the five Yayu cooperatives would be around 250 kg/ha (437.5 kg per farm), which would be similar to average yields quoted for Ethiopia, for example, 268 kg/ha (Technoserve & Sustainable Trade Initiative, 2013), close to reported minimal yields, for example, 260 kg (Global Coffee Platform, 2016), but below other reports for average yields (Global Coffee Platform, 2017).

As we have mentioned, global coffee prices are susceptible to extreme volatility. In the 2018/19 harvest, prices for coffee were high in Ethiopia, with the price of dried cherry doubling to 30 ETB/kg (Gole pers observ.) from 15 ETB/KG in 2017/18. However, this was of little consolation to many Ethiopian farmers, including those at Yayu, as the harvest in 2018/19 was very poor (e.g., 0–200 kg/ha, at Yayu) due to lack of rain during critical periods in the coffee growing cycle (see above) resulting in dramatic shortfalls in coffee income.

Of the 115 households (out of 272) that received an extra set of questions on farm management (see Supplemental Files), the following was reported: none of them used chemical fertilizers; the youngest coffee trees on their farms were 11 years old, the oldest were on average 23 years old, and some up to 30 years old; 97% of the respondents indicated the use of family labor for farm management (e.g., weeding); 67% indicated the use of hired labor (for weeding, harvesting and processing). The common assumption that smallholders only use family labor is therefore not true for our case study.

All of the five primary cooperatives in our survey purchased processing equipment to mechanically wash (at least a proportion of) their coffees. The washing machines were purchased through banks loans, facilitated by the non-profit organization Technoserve. Access to this equipment introduced the cooperatives to higher value washed coffees, compared to the lower quality (and lower value) natural (sundried) processed coffees previously prepared by the farmers. It should be noted, however, that farmers in Ethiopia can, and do, produce specialty coffee via the sundried method, although the greater proportion of specialty coffee is provided by washing (Technoserve, 2016). Our study enhanced the available resources at Yayu, by providing training in harvest and post-harvest processing, establishing protocols for these activities; and by providing key materials needed to harvest and process specialty coffee, including the provision of raised coffee drying beds (African beds). Our case study also afforded access to a specialty coffee buyer (Union Hand-Roasted Coffee, London, UK) who was willing to pay a premium for the coffee, if it scored 84 points or above on the Specialty Coffee Association (SCA) scale (see “Methods”). The coffee in this study received a specialty cupping score ranging from 84 to 86, and was sold as Yayu Forest Coffee.

The farmers sold either fresh cherry or sundried cherry. Middlemen received both fresh and sundried cherry at the time of the survey, whereas the five cooperatives only received fresh cherry for washing (none of the cooperatives were accepting sundried cherry). For washed (specialty) coffee, the farmers delivered fresh cherry to one of the five primary cooperatives, where it was processed using the washed coffee method to produce parchment coffee (see “Methods”). The secondary cooperative (union) then transported the parchment coffee to Addis Ababa, where it was further processed, including the removal of husk and beans with faults (defects), prior to export. The husk formed approximately 20% of the parchment weight; that is, with a conversion ratio of 1.25:1 as per international standards (International Trade Centre, 2011). The percentage of rejected coffee beans depended on the number of defects (see “Methods”).

During the survey we determined how much each farmer sold through each of three sales channel. (1) Sundried cherry. Collectively, the famers in the survey sold a total of 134,668 kg of dried sundried cherry; the average amount sold per farm was 524 kg. Sundried cherry was sold exclusively to middlemen. (2) Fresh and sundried cherry. Farmers sold both fresh and sundried cherry; 6% sold only fresh cherry (and no sundried cherry) and 10% sold only sundried cherry (and no fresh cherry). Fresh cherry is sold the same day to the cooperative or a middleman. Sundried cherry is dried at home, taking 20–30 days to dry before being sold (to middlemen), with drying either undertaken on the ground (on the earth or a plastic) or on a raised (African) drying bed. Sundried cherry is sometimes also used for home-consumption by the producing household. (3) Fresh cherry for washing. Collectively, the farmers in the survey sold a total of 137,286 kg of fresh cherry for washing (for specialty coffee); with an average amount per farm of 534 kg.

In our survey, across the five cooperatives, the requirements for the above three production volumes required 541,242 kg of fresh cherry, with an average of 2,106 kg per farm, which is equivalent to 702 kg of sundried cherry, or 351 kg of clean coffee (GBE). This corresponds very closely to the calculation of 200 kg per ha, or 350 kg per average farm (at 1.75 ha). These three sales channels were used for the calculating farm gate prices and farmer income from coffee (see below).

Coffee price structure and the value chain

In our survey, the FOB price was 2.80 $/lb (6.16 $/kg), which included a 0.20 $/lb forest conservation premium, paid by the FOB purchaser (i.e., the importer/roaster). In this case study, the Direct Trade mechanism was employed, whereby the producers/farmers were in a close relationship with the purchaser, with access to the FOB price and details of the forest conservation premium. An FOB price of 2.80 $/lb is twice the Fairtrade minimum price of 1.40 $/lb paid for Arabica coffee. In the period 2016, 2017, 2018 the world market FOB price for clean (exportable) coffee, classified as “other milds”, was 1.63 $/lb, 1.50 $/lb and 1.32 $/lb, respectively. The price for sundried naturals (Brazilian naturals) was 1.37 $/lb, 1.31 $/lb and 1.14 $/lb (International Coffee Organization, 2020), respectively. The International Coffee Organization (International Coffee Organization, 2020) also reports on prices paid to growers for clean coffee (GBE) by country: in 2017, the average price paid to growers in Ethiopia for exportable (green) coffee was 0.82 $/lb (1.80 $/kg). Based on a 6:1 conversion this would be a value of 0.30 $/kg (6.9 ETB/kg) for fresh cherry. By producing high-quality, washed (specialty) coffee the farmers in this case-study were thus receiving a significant increase in value for their coffee, considerably above: national average prices; prices including premiums paid through certification; and world market (commodity) prices. However, it should be said that some certification systems bring specific welfare benefits, which were not included in our study.

Unlike traditional middlemen (akrabi), or private coffee washing stations, the secondary cooperative (the union) and the five primary cooperatives are owned by its members, as a business. The ultimate authority in a cooperative is the General Assembly, in other words, the total ownership of the business. In Yayu, at the end of each fiscal year the five primary cooperatives hold an annual general meeting (AGM) with its General Assembly to discuss the production and sales outcomes from each production year. The primary cooperative members choose what to do with the profit made from coffee sales. In Ethiopia, a rule is that 70% of the profits of the primary cooperatives need to be transferred to the farmers, while the rest is kept for functioning of the primary cooperative.

At the farm gate level, data on the price paid per kg fresh cherry and dried cherry were obtained from the FGD and key informant interviews (see Supplemental Files). The price of sundried cherry was 0.65 $/kg (15 ETB/kg), with a fresh cherry to sundried cherry conversion is 3:1. This means that when fresh cherry has a price of 0.21$/kg (5 ETB/kg) or higher it is more profitable, and requires less input (e.g., drying and storage), to sell fresh cherry as opposed to sundried cherry. For example, a farmer that sells 1 kg of sundried cherry (requiring 3 kg of fresh cherry) at 0.65 $/kg (15 ETB/kg), can sell 3 kg of fresh cherry (at 0.48 $/kg (11 ETB/kg) for 1.43 $/kg (33 ETB/kg)), representing an increase of 120%. However, the fresh cherry must be delivered to the primary cooperative (for washing) within 24 h, which imposes logistic issues and perhaps costs.

The breakdown of the pre-import vale chain for our case study is as follows. The buyer (the importer) pays an FOB price of 2.80 $/lb, of which 0.26 $/lb is collected in costs/deductions by the secondary cooperative (the Union). Based on an average price of 0.48 $/kg (11 ETB/kg) for fresh cherry, with a conversion of 6:1 (0.47 $/kg × 6/2.2) the cost of coffee (i.e., fresh cherry) purchased by the primary cooperative is 1.30 $/lb (46.4% of the FOB price). A cost of 1.24 $/lb (44.3% of FOB price) is deducted by the primary cooperative, for their costs, which include transport, labor, washing station costs (e.g., fuel, maintenance), drying bed materials, coffee sacks, lost revenue (due to rejected coffee beans), and interest on loans (needed to pay the farmer for their fresh cherry). The largest cost the primary cooperative is that of purchasing fresh cherry from the farmers (1.30 $/lb), which is also equivalent to the farmer’s income from coffee. The 1.30 $/lb received by the farmers (the farm gate price) includes the 0.20 $/lb premium and the share of the profits post export. These relationships are summarized in Table 1. It is important to note that the level of specialty coffee varies between years, and can vary from year to year, depending on a variety of influences (e.g., coffee price, weather, inter-annual production levels); in our case study demand was not a limiting factor.

Table 1 Breakdown of main elements in case study supply chain for specialty (washed) coffee, from farm to export.

US$, US Dollar. Exchange rate of 1 US$ to 23 ETB. FOB price includes 0.20 US$/lb conservation premium. Bold indicates key total values.

Value chain actor	Main responsibility	Major cost(s)	Cost (US$/lb)	Coffee stage	Percentage of total cost ($/lb) (%)	
Farmer	Owns farm; grows and harvests coffee	Seasonal labor (weeding, harvesting)	Not recorded	Fresh cherry	N.A.	
Primary cooperative	Owns wet mill (for washing); buys, washes and stores coffee	Cost of coffee (paid to farmer)	$1.30	Fresh cherry	46.4	
Primary cooperative	As above	Operating costs: labor, fuel, finance, administration, maintenance	$1.24	Fresh cherry to parchment	44.3	
Secondary cooperative (union)	Transportation and export	Operating costs: dry-mill rental, transport, export logistics, administration	$0.26	Parchment to clean (green) coffee	9.3	
		(Total) FOB price	$2.80		100	

All cooperatives had the same export costs, amounting to 9% of the FOB price. As we reported above, the farmers in our survey received on average 0.48 $/kg (11 ETB/kg) for fresh cherry, but this varied, from 0.32–0.62 $/kg (7.3–14.33 ETB/kg) across the five cooperatives, 32% less, and 32% more, than the average. Thus, in some cases, the higher FOB price translated into a higher farm gate price, and in other cases it did not, depending on a number of factors linked to cooperative management and efficiency. During the focus group discussions, it became clear that the differences in the prices paid for fresh cherry (by the primary cooperative) were due to the cost-structures within each cooperative (e.g., for seasonal labor hired) and other financial considerations.

Amongst the most important financial considerations linked to the proportion of the FOB price received by the farmers, are the conversion rates/ratios. In order to meet the standards of the specialty coffee market the conversion rate (ratio of fresh cherry to clean coffee) often can be higher than 6:1. The conversion rate covers all the usual factors, especially weight loss (due to loss of water, removal of mucilage and parchment), but also increases with the increase in quantity of rejected coffee beans (due to defects; see “Methods”). The defective beans are usually sold at the local market at a lower price, and so some costs are recuperated, but any gains from this revenue channel are greatly exceeded by the higher conversion rates, which would have a big influence on farm gate income and profitability. All of the five cooperatives in our case study had a final (post farm gate) conversion rate of 8:1 or higher (up to 11:1), when producing specialty coffee. These higher conversion rates thus result in farmers receiving a lower percentage of the final FOB price. This could not be quantified in our study because none of the cooperatives had either sound bookkeeping practices or optimal management procedures to measure the key variables. Consequently, the primary cooperatives have little understanding of how the purchase of lower quality cherry (for washed, specialty coffee) at higher prices (i.e., 1.30 $/lb) affects the profitability of the cooperative, due the higher conversion rates from fresh cherry to clean specialty coffee. Limited financial understanding also caused the primary cooperatives to take out large loans as working capital, which resulted in paying an elevated amount of interest, reducing profits and therefore household income.

Farmer income from coffee

Revenue per hectare (ha) is a function of yield levels (which vary each year) and the market price obtained per unit of output. Accordingly, revenues differ as yield changes even if the coffee price changes, and vice versa. In this case study we used averaged figures across our farmer study group and the five Yayu cooperatives, for the harvest season 2017/18, which was an “off” year in the biennial production cycle. The farmers harvested on average 2,106 kg of fresh cherry, which gave 702 kg of sundried cherry, and 351 kg clean coffee (Table 2).

Table 2 Calculation of earnings from coffee based on survey data, under three scenarios.

(1) Baseline scenario (non-specialty coffee, total harvest used to produce sundried coffee); (2) Improved scenario, where 75% of harvest is used to produce sundried coffee and 25% goes into specialty (via the washed coffee processing method), which represents the current mode for the five Yayu cooperatives; and 100% specialty scenario, where all the harvest goes to specialty/washed coffee. *Under all scenarios 351 kg of coffee is produced, from 2,106 kg of fresh cherry. Clean coffee produced via the sundried route, requires a twostep process Fresh cherry to sundried conversion is at ratio of c. 3:1 (i.e., 2,106 kg/3 = 702; and conversion of sundried cherry to clean coffee is at a ratio of 2:1 (i.e., 707/1 = 351 kg). The overall conversion rate (ratio), from fresh cherry to clean coffee is thus is 6:1. ETB, Ethiopian Birr; US$, US Dollar. Exchange rate of 1 US$ to 23 ETB. Bold indicates key total values.

Scenario	Sundried/fresh cherry requirements (kg)	Fresh cherry required (kg)	Total coffee (kg)	Cost per kg sundried/fresh cherry (US$/ETB)	Coffee income on sale of 351 kg coffee (ETB)	Coffee income on sale of 351 kg coffee (US$)	Percentage Increase on Baseline Scenario (%)	
Baseline scenario, all sundried	702	2,106*	351	0.65/15	10,530	457.83		
								
25 % specialty scenario, part washed, part sundried								
Sundried cherry	524	1,572	262	0.65/15	7,860	341.74		
Fresh cherry	534	534	89	0.48/11	5,874	255.39		
	Total	2,106	351		13,734	597.13	30.43	
								
100% specialty scenario, all washed	2,106	2,106	351	11	23,166	1,007.22	119.99	

The three main coffee production scenarios for the study area serve to provide a means of understanding the relationship between farmer income and the type and scale of the coffee production, and the influence of specialty coffee prices.

Baseline scenario—non-specialty coffee. Prior to this case study (pre 2014), that is, before the five Yayu cooperatives had access to the specialty coffee market, farmers would process all of their cherry in the traditional way, selling it as sundried (natural) coffee. The average annual gross income per household across the five cooperatives for processing and selling coffee in this manner would be US$ 458 per year/season. This is based on converting 2,106 kg cherry to 702 kg of sundried cherry (at a conversion rate of 3:1), with 1 kg of dried cherry selling at 0.65 $/kg (15 ETB). See Table 2.

Improved scenario—25% specialty coffee. In the main survey period (2017/2018 harvest) farmers had the option to participate in washed coffee processing for specialty coffee. On average farmers sold 25% of their harvest as fresh cherry (534 kg) for washed/specialty coffee and 75% as sundried cherry (524 kg, requiring 1,572 kg fresh cherry) for the traditional market channel. Total income from selling a combination of sundried cherry and fresh cherry was US$ 597, with a mixture of prices for each: either 0.65 $/kg (15 ETB/kg) for sundried; or 0.48 $/kg (11 ETB/kg) for fresh cherry for washing (Table 2). This represents an income increase of 30.4 % above the (non-specialty coffee) Baseline Scenario of selling only sundried cherry (income per year/season US$ 458).

100% specialty coffee scenario. This is a hypothetical but workable scenario. If farmers were to sell all (2,106 kg) of their fresh cherry at 0.48 $/kg (11 ETB/kg) for washed (specialty) coffee, through the primary and the secondary cooperative, they would earn US$ 1,007 (Table 2). This scenario represents a value increase of 69% above the Improved scenario – 25% specialty coffee and 120% above the Baseline scenario—non-specialty coffee.

Under both the Baseline scenario —non-specialty coffee and Improved scenario—25% specialty coffee, income from coffee does not allow the farmers of the five cooperatives to exceed the official poverty line of US$ 1.90/day or US$ 693 per year (Ferreira, Jolliffe & Pprydz, 2015), unless they were able to provide additional (non-coffee) income of $235 (an additional 33%) and $96 (and additional 14 %), respectively. Coffee income in an “on” year in the biennial production cycle, with an estimated 25% increase in productivity and income, is also unlikely to see the poverty line exceeded under the Baseline scenario, but it could be exceeded under the 25% specialty scenario. For example, if the farmers were to attain our estimated average “on” year yield of 440 kg per farm (see above for estimate details) then coffee income would increase to $572 and $746, respectively. Farmers would only consistently exceed the poverty line (based on coffee income alone) if they would sell all their coffee under the 100% specialty coffee scenario, that is, an income of US$ 1,007 (or $1,259 in an “on” year) via all coffee sold as fresh cherry to the secondary cooperative (to be washed) for specialty coffee.

Our survey did not specifically ask why the farmers sold only a small percentage of their cherry (i.e., their harvest) to the secondary cooperative for washed (specialty) coffee. However, during qualitative interviews with cooperative management, the farmers of the five cooperatives gave a number of reasons, including: access to, and timing of, cherry purchase by the primary cooperative, which of course has a direct bearing on the timing of income; distance to the collecting points (particularly when harvest quantities were low), and the convenience (and logistics) of selling sundried cherry vs. fresh cherry. Sundried cherry has the potential to be stored for many months, whereas the latter has to reach the washing station within c. 24 h.

Spatial analysis of forest and agricultural zones

We produced a simplified classification of Landsat EVI 2017 values for the Yayu Coffee Forest Biosphere Reserve area, including: (1) grass/scrubland/urban, (2) crops/agriculture, (3) forest/transition forest, (4) the Yayu agricultural zone; and (5) our coffee farm data (GPS locations), superimposed on the (i) core area, (ii) (forested) buffer zone and (iii) transition zone, of the Yayu Coffee Forest Biosphere. The spatial relationship between the classifications and the reserve boundaries is shown in Fig. 1. We found that most of the coffee farms were located in the (forested) buffer and forest transition zones (Fig. 1). Mean Landsat EVI 2017 values for the Yayu Coffee Forest Biosphere area/zones are as follows: (1) core area (0.58); (2) buffer zone (0.55); (3) transition zone (0.46); (4) the Yayu agriculture zone (0.37); and (5) our coffee farm data (0.50), as shown in Fig. 2 and Table 3. The analysis reveals that 92% of coffee farm locations have an EVI of over 0.4 and are located in forest with a spectral quality approaching the (forested) buffer zone, and as good as the best of the transition zone forest, and considerably better the Yayu agriculture zone (Fig. 2). The P-values demonstrate that coffee farms are not a random sample (Table 3).

Table 3 EVI and forest cover values for Yayu Coffee Forest Biosphere Reserve and coffee farms.

EVI analysis values (mean, median, standard deviation, number of samples and P-values) and percentage forest cover. See Figs. 1 and 2, and “Methods”.

Zones/coffee farms	EVI values	Forest cover (%)	
	Mean	Median	Standard deviation	Number of samples	P values compared to coffee farms		
Core area	0.58	0.58	0.10	319,455	<2.2e−16	99	
Buffer zone	0.55	0.56	0.10	236,819	<2.2e−16	96	
Coffee farms	0.50	0.51	0.08	340	1	90	
Transition Zone	0.46	0.46	0.11	1,327,713	<2.2e−16	78	
Yayu, Agricultural Zone	0.37	0.37	0.11	43,618	<2.2e−16	55	

Forest cover levels (Global Forest Watch, 2014) for the same five classifications as the EVI analysis, ranged from 55% to 99%: core area = 99% forest cover; buffer zone = 96% forest cover; coffee farm locations = 90% forest cover, transition zone = 78% forest cover, and Yayu agricultural zone = 55% cover (Table 3). Deforestation levels for the Yayu area (as circumscribed in Fig. 1) are extremely low (a few pixels) and often insignificant. The transition zone shows the most deforestation with a 1.4% loss since 2001. Deforestation for the coffee locations was extremely low (a few hectares (15ha) between 2001 and 2018). However, this measurement cannot to be relied on, because at this resolution and scale (1 ha per year) any differences are likely due to noise in the data (e.g., very small scale land use change, or to non-vegetation factors, such as cloud cover). Reforestation was also at insignificants levels, that is, less 0.1 ha.

Discussion

Increasing farmer income through coffee production via specialty coffee

To our knowledge, this is the first available study quantifying the income from coffee earned by small-scale coffee producers in Ethiopia. Calculating the income from coffee for small holder farms in Ethiopia, and understanding the pre-export value chain, was a complicated undertaking. Regardless, it was necessary to do this, to understand the benefits and disadvantages of using specialty coffee production to increase income, and to understand the potential benefits for biodiversity conservation.

Most smallholder coffee farmers in the Illubabor zone of southwestern Ethiopia have been locked out of the global specialty coffee supply chains, due to the production of poor coffee quality and lack of access to markets (Technoserve, 2016). Higher quality coffee from Illubabor is currently sold as generic Limu commodity coffee (Davis et al., 2018; ECX Coffee Contracts, 2015) resulting in a loss of uniqueness and market differentiation, and thus the potential for higher prices. The price of commodity coffee is set by the C-market, which is plagued by volatility (International Coffee Organization, 2020). In recent years (2014–2020), market conditions have set prices at very low levels (International Coffee Organization, 2020), with the result that many of the world’s coffee farmers cannot cover the costs of production, resulting in a net income loss (Montagnon, 2017). Specialty coffee can provide an alternative market system, which is not directly linked to the commodity market, whereby producers (farmers and cooperatives) who produce high-quality coffee can access independent pricing structures.

In our case study we show that even moderate participation in the specialty coffee market can positively influence income for the smallholder farmer. We show that farmers can increase their income from coffee (representing 70–90% of household income) by up to 30%, by selling a proportion (c. 25%) of their harvested coffee into the specialty coffee sector, compared to a pre case study baseline (Baseline scenario—non-specialty). In our 100% specialty scenario, where 100% of production of fresh cherry would be sold for washed (specialty) coffee, farmers could increase their income from coffee by 120%, although this would require optimum operating efficiency of the cooperatives and their farms. Participation in the specialty coffee sector can thus produce a substantially improved coffee income, well above that achievable within a system that uses market value (commodity) coffee and certification. In our case study the FOB price was 2.80 $/lb. From 2015 to 2018, a total of $1,258,165 (£924,751) of specialty coffee was purchased across the five cooperatives, much of this constituting additional revenue, although this income was not evenly distributed across the five cooperatives.

Despite the demonstration of positive income benefits for producers via engagement in specialty coffee, we also identified some key caveats, which would be applicable across much of Ethiopia and also many other coffee producing countries. It is clear that enhanced income from specialty coffee relies on much more than coffee quality.

Caveat (1)—management

To achieve the best outcomes via specialty coffee (e.g., our 100% specialty scenario), requires near-optimum operating performance of the farm, the primary cooperative, and the secondary cooperative (the union). In particular, the proportion of the increased coffee income going back to the farmers requires the management activities of the primary cooperatives to be efficient and effective (see below for further details on cooperative management). A key issue is the selling of coffee to middlemen, in order to improve income security via earlier payments. That the farm gate prices (i.e., price paid for fresh cherry) differed between cooperatives (even when cooperatives receive the same FOB price) is important, as it shows the limitations of a strategy that focusses only on paying farmers a higher price in order to increase income. Further details on cooperative management are given below.

Caveat (2)—conversion rates

Many of the costs for specialty coffee are the same as that for producing green coffee via the sundried route (e.g., labor costs, weeding and harvesting), and the overall coffee conversion ratios up to the farm gate level can be the same (e.g., 2,106 kg fresh cherry converts to 315 kg clean coffee under our three different pathways). The final conversion rate, however, was much higher for specialty coffee (e.g., 8:1 and up to 11:1), due to rejection of defects (coffee beans with faults), especially at the later stages of processing (e.g., milling, grading and sorting). Losses in the total amount of exportable clean coffee is deducted from the FOB price and hence cooperative and farmer profits, although we were unable to quantify this in our study (see “Results”).

Caveat (3)—investment

Initial and ongoing financial investments (for farmers and their cooperatives) are required to enter and maintain a position in the specialty coffee market. In terms of the specialty coffee, purchase or access to washing stations (wet mills) is particularly pertinent (Baptista & Jenkins, 2017; IPE Triple Line, 2017; Minten et al., 2019; Technoserve, 2016), as the larger share of the specialty market requires washed coffee. In this study we did not factor-in the initial cost of the washing stations, other processing equipment, and training, although we were able to observe that over the 4-year period of the case study (see above) the Yayu cooperatives were able to repay the loans for the washing stations, and that the costs associated with the investment required to enter the specialty market was recouped, due to the increase in quantities of coffee purchased, and improved farm gate price and (FOB) price of the coffee purchased.

Caveat (4)—direct trade

In this case study the Direct Trade model was employed, to ensure that the benefits of participation in the specialty coffee sector reached the producers. The following interventions were required to make this approach work: all members of the value chain were familiar with key metrics, for example, farm gate price, FOB, and retail value, and the conservation premium (0.20 $/lb); the purchaser (i.e., the importer) conducted annual site visits to Yayu, and Addis Ababa, to check that key operating and accounting procedures were in place and working, and in particular that the producers (farmers) were in receipt of the correct farm gate price. In this way it was possible to avoid the main failings detected in problematic Direct Trade relationships (MacGregor, Ramasar & Nicholas, 2017; Vicol et al., 2018), as outlined in the “Introduction”.

Caveat (5)—external factors: prices and other factors

Revenue per hectare (US$/ha) is a function of yield levels and the market price obtained per unit of output. Even when (FOB) prices are set above commodity market prices, income may fall below expectations due to numerous other factors, including biennial bearing differences, which can be particularly marked in Ethiopia (Davis et al., 2018), pest and diseases outbreaks, extreme weather events, and climatic variability (Moat et al., 2017). These factors may be exacerbated by: limited farm management, the low level of soil fertility inputs, lack of pest and disease control, absent or limited composting or mulching, and a stumping cycle that goes beyond recommended timings (Davis et al., 2018), despite the tangible benefits (IPE Triple Line, 2017; Technoserve, 2016). When global commodity coffee prices spike, the proportional gains of higher-priced (and especially fixed-price) specialty coffees are reduced.

Cooperative management

Of the above five caveats, cooperative management is identified as one of the most important, and perhaps the most easily actionable caveats. Indeed, the significance of strengthening the management and governance of coffee cooperatives in Ethiopia has been highlighted by others (Ruben & Hoebink, 2015; Woubie, Muradian & Ruben, 2015; Baptista & Jenkins, 2017). Primary cooperative activities can play an effective role in supporting coffee farmers by supplying an effective management system, capital, and transportation and other logistics, which individual small-scale Ethiopian farmers lack. In addition, the collective cooperatives and secondary cooperatives can provide coffee farmers with negotiation powers in the international market, compared to the individual farmer (Kodama, 2007).

In order to sustain and enhance the performance of the primary cooperatives at Yayu, we identified specific actions for improvements, including: (1) more detailed and more consistent financial management and record keeping; (2) a more workable distribution in the timing and reliability of coffee income (e.g., by providing income security to reduce the production of (lower value) dried cherry, or side-sales of fresh cherry outside the cooperatives); (3) improving harvest procedures (especially the picking of higher proportions of mature (rather than unripe cherry)); (4) improvements in post-harvest processes (e.g., rejection of low quality cherry at time of purchasing, and optimizing depulper calibrations) to improve the conversion ratio of fresh cherry to clean dried coffee; (5) dealing with logistic bottlenecks and low-volume situations during harvest, for example, more efficient systems for delivering small amounts of cherry to the primary cooperatives (and their washing stations), and during other production stages. Further training, to improve cooperative management (including farmer financing structures), and application of technologies (e.g., use of computers, and mobile phones) would facilitate the resolution of many of these issues.

This case-study highlighted how poor organizational structure can influence the transfer of benefits from price premiums, and other benefits, to smallholders (Minten et al., 2019; Vicol et al., 2018). Similar conclusions have been made for other coffee cooperatives in Ethiopia (Ruben & Heras, 2012; Ruben & Hoebink, 2015; Woubie, Muradian & Ruben, 2015), but there are also other parts of the value chain to consider, including the governance of certification systems (Jena et al., 2012), and national trading mechanisms in Ethiopia (Worako et al., 2008). These factors are applicable not only to cooperatives and the value chain across Ethiopia but also further afield.

Specialty coffee and biodiversity conservation

Our EVI analysis shows that the spectral quality of the forest canopy for the 340 coffee farm locations are comparable to, and mostly better than, the Yayu coffee forest biosphere transition zone, and much better that the agricultural zone (Fig. 2). Our analysis of forest cover (Global Forest Watch, 2014) shows that the coffee farm locations attain 90% forest cover, compared to 99% for the core area, and much more favorable than the Yayu agricultural zone (55%), and the Yayu reserve transition zone (78%). These two groups of metrics are congruent (Table 3) and when combined provide a useful method for initial and remote assessment of forest (canopy) health and cover (area). Moreover, we were able to show that specialty coffee, and the increase in income that it can bring, can be linked to forest, and, by proxy, biodiversity preservation. It should be carefully noted, however, that the biodiversity value of forest coffee systems in Ethiopia, although considerable, is not the same as undisturbed forest areas (Gole, 2003; Gole et al., 2008; Gove et al., 2008; Hylander et al., 2013), and that the undisturbed core zones of protected areas, such as that at Yayu, should be (and need to be) maintained (Moat, Gole & Davis, 2019). In addition to remote measurement of the environment, that is, via satellite data, direct quantification of biodiversity on the ground, by measurement of its constituent elements (e.g., animals, plants and fungi) is still required.

In our study the FOB price of the Yayu (specialty) coffee included a 0.20 $/lb forest conservation premium, as a means of consolidating the link between the coffee and the environment, and as part of a mainstreaming mechanism for forest conservation via coffee trading. It was clear, from the workshops and numerous anecdotal conversations, that Yayu coffee farmers were very much aware of the intrinsic value of the forest, but that if coffee farming were to become unsustainable (unprofitable) then options for livelihood sustainability would undoubtedly include the removal of forest cover for other agricultural activities, outside of the core area of the Yayu Coffee Forest Biosphere Reserve. This would have an extremely negative influence on forest cover and biodiversity. The conservation premium alone (0.20 $/lb) would not be sufficient to improve household income, or drive land-use decision making, but it served to reinforce the message to value chain actors, and especially farmers and consumers, that added value via specialty coffee can be linked to forest and biodiversity conservation. The aim is that both the elevated prices obtained (via specialty) and the conservation premium will serve to influence biodiversity-positive land use decisions, that is, the retention of forest cover. It should also be emphasized that the increases in income achieved via specialty coffee reported here were achieved without the need for more land, or increased (biodiversity negative, and carbon positive) inputs, such as artificial fertilizers, irrigation, or herbicides and pesticides. It is fortunate that coffee production in Ethiopia relies on suitable forest canopy cover (Davis et al., 2018; Moat et al., 2017). More generally, we suggest that coffee production, or products directly linked to environmental benefit, should have suitable and transparent environmental metrics, either via techniques similar to those used here, or via on-the-ground biodiversity assessment, and preferably both. The main advantages of using satellite technology is that they are cost effective, expedient and time-bound.

Supplemental Information

Supplemental Information 1 Yayu farmer survey dataset.

Click here for additional data file.

Supplemental Information 2 Yayu farmer survey.

Survey questions for Yayu primary cooperatives, in English.

Click here for additional data file.

Supplemental Information 3 Yayu Farmer’s Survey.

Questionnaire in Amharic (blank form).

Click here for additional data file.

Supplemental Information 4 Yayu farmer survey, additional questions.

Click here for additional data file.

We would like to thank the Amar-Franses and Foster-Jenkins Trust for supporting fieldwork activities in Ethiopia. We gratefully acknowledge the coffee-farming community at Yayu for their participation in this project, and especially for their participation in the main survey and focus group discussions.

Additional Information and Declarations

Competing Interests

Author Contributions

Human Ethics

Data Availability

Pascale Schuit, Jeremy Torz & Steven Macatonia are employed by Union Hand-Roasted Coffee (UHRC). Graciano Cruz is employed by HiU Coffee.

UHRC and HiU coffee were official industrial partners of the Darwin Initiative (UK; DFiD) project, ref. no: 2 2-006

Pascale Schuit conceived and designed the experiments, performed the experiments, analyzed the data, prepared figures and/or tables, authored or reviewed drafts of the paper, and approved the final draft.

Justin Moat conceived and designed the experiments, performed the experiments, analyzed the data, prepared figures and/or tables, authored or reviewed drafts of the paper, and approved the final draft.

Tadesse Woldemariam Gole conceived and designed the experiments, authored or reviewed drafts of the paper, and approved the final draft.

Zeleke Kebebew Challa conceived and designed the experiments, performed the experiments, prepared figures and/or tables, authored or reviewed drafts of the paper, and approved the final draft.

Jeremy Torz conceived and designed the experiments, performed the experiments, authored or reviewed drafts of the paper, and approved the final draft.

Steven Macatonia conceived and designed the experiments, performed the experiments, authored or reviewed drafts of the paper, and approved the final draft.

Graciano Cruz conceived and designed the experiments, performed the experiments, authored or reviewed drafts of the paper, undertook quality improvement, and approved the final draft.

Aaron P. Davis conceived and designed the experiments, performed the experiments, analyzed the data, prepared figures and/or tables, authored or reviewed drafts of the paper, and approved the final draft.

The following information was supplied relating to ethical approvals (i.e., approving body and any reference numbers):

The survey did not go through an institutional review board. The Royal Botanic Gardens, Kew conforms to due diligence imposed by the host country and the grant awarding body. When seeking ethical consent in Ethiopia, we were advised by Ethiopian Governmental officials to consult the coffee farming cooperatives leaders (local governance for cooperative members), which is what we did.

The following information was supplied regarding data availability:

Raw data are available in the Supplemental Files.

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
