# Peer review of "The potential for income improvement and biodiversity conservation via specialty coffee in Ethiopia"

_PeerJ, doi:10.7717/peerj.10621_

## Round 0.1 · original submission · Minor Revisions

As you can see from the reviews included, the two reviewers I have consulted are positive and generally support the publication of the manuscript after some minor comments are addressed.

Personally, I happen to disagree with Reviewer #1's suggestion to move some of the text related to value chain and economic calculations to an appendix: in my opinion, such aspects are central to the work and should be retained in the main text. Having said that, however, I encourage you to try and shorten those sections, if possible, and to make them more accessible to non-economists by reducing or avoiding as much as possible economics jargon.

I congratulate you on this interesting and valuable research and look froward to receiving the revised version of the manuscript.

·

Basic reporting

The present case study gives a very detailed account of the coffee value chain in the Yayu area of Ethiopia and highlights potential income improvement and biodiversity conservation via specialty coffee. The paper is written in professional English and pleasant to read. It contributes to a better understanding of farmer decisions along the coffee value chain and possible interventions for income improvement. However, it is rather lengthy and I encourage the authors to shorten text that dilutes the focus of the paper, for instance the review of coffee certification schemes in the introduction and repetitions in the Methods section (line 294ff) and results (lines 622ff). Moreover, as an ecologist, I found the detailed description of the value chain and economic calculations quite hard to digest. Given that PeerJ has a focus on the biological and environmental sciences I would recommend shortening those parts to provide a more balanced account of economic and biodiversity issues. Perhaps some of the information could go into an appendix.
Generally, I find that the term “biodiversity” is used in a very generic way. The authors should be clearer about when they refer to forest cover (as a proxy indicator of biodiversity) and when they refer to actual biodiversity indicators such as genetic diversity, species diversity or ecosystem diversity. There is a bulk of literature from Ethiopia and other countries that shows that tropical forest degradation cannot be detected using canopy cover as the only indicator. Thus, the findings of the study with regard to biodiversity are rather preliminary.
Another concern of mine is that the manuscript does not sufficiently highlight that Ethiopia is the great exception to all coffee countries because Arabica coffee originates from Ethiopia and much of the forest coffee in southwestern Ethiopia is managed in its natural habitat. How do results from this particular case study apply to other types of coffee growing areas in Ethiopia, and worldwide, in particular with regard to biodiversity conservation?
Finally, I may add that the first author is employed by the company that purchased the specialty coffee from the case study area and I wonder if this may not create a conflict of interest.

Experimental design

The data collection method is generally well described except for a couple of shortcomings that I would like to point out. The average reader will not be familiar with the terms Forest Coffee, Semi-forest coffee and Forest Garden Coffee and thus they should be clearly defined (lines 183-184). I wonder if coffee at Yayu is really produced in Forest Coffee systems because later planted coffee (line 412) as well as farms and farm management (line 435) are mentioned. Lines 262ff: It is really hard to follow the description of the different parts of the coffee bean. How about an illustration? Is skin (pericarp) = hard coating (parchment)? Later parchment coffee is mentioned, which should be described here, too.
Fig. 1 is hard to decipher; besides, the buffer zones do not seem to be indicated in the map. An inset overview map of Ethiopia would be helpful.

Validity of the findings

The results part is very lengthy and descriptive. I wonder if some of the information could better got into the methods part, an annex, or be presented in a figure or table. The authors should take greater care to directly link the results to the three research questions; this holds true for Section 3.1 in particular.
Section 3.2 contains a mixture of results and discussion, which I would strictly separate (e.g., lines 500 ff). Furthermore, I recommend using a table or flow chart for the results rather than stating so many figures in the text which is quite a heavy read.
In Section 3.3, I have some questions about the scenarios: Is “specialty” determined each year? So could it be that in one year, the coffee scores lower and farmers will not be able to sell it as specialty coffee? Is it realistic that farmers can sell 100% of their coffee as specialty coffee, considering the logistic constraints (limited transport time)? Is the market large enough to absorb 100% specialty coffee?
Similar to the results, there is a need to double check the discussion against the three research questions stated in the beginning. I miss a more thorough discussion of whether this case study is applicable to other parts of Ethiopia, considering coffee quality, logistics and forest cover. How representative is it in terms of other coffee growing areas worldwide?
More importantly, the authors need to be more cautious regarding the links between biodiversity conservation and specialty coffee. In line 770, the term “quality of the forest” is used, but I do not think that a simple forest cover analysis can tell much about forest quality. Fig. 2 comes without any statistical test on the difference between the EVI values between the different forest types. Last but not least, the authors note themselves that the biodiversity value of forest coffee systems is not at the same level as undisturbed forest. Considering that the paper is equally about income improvement and biodiversity conservation, this warrants a more thorough discussion that should also take into account the special situation in southwestern Ethiopia. For instance, in eastern Ethiopia (Harar) coffee is grown with less shade and irrigation may be necessary.
Table 1: The conservation premium should be indicated in the table. Is this premium paid to the farmers in any case, or only for specialty coffee?
Table 2: I could not make sense of the second and third column in the table.

Additional comments

Introduction: The introduction claims that biodiversity loss and poverty are usually linked. Well, agricultural intensification and wealth can also be linked to biodiversity loss as we know from developed countries. So I would encourage the author to develop their argument in a more convincing manner, using a variety of more recent literature resources (they rely heavily on Scherr & McNeely 2007). For instance, there is a bulk of literature on the land sharing vs. land sparing topic. In line 80, “eco-agricultural (or agroecosystem) coffee production” is mentioned, but not defined. It is not clear to me if this expression refers to the special situation in southwestern Ethiopia, or to different types of global coffee production systems. In line 132, the term specialty coffee is introduced but only defined later in lines 142ff.

·

Basic reporting

Very well written and structured. Still some small errors

Experimental design

Well researched. But some questions:
1. All cooperatives are Rainforest Alliance certified. Please explain what this means for a. coffee prices and b. other benefits to the cooperatives and its members.Please bring this also down in sections 3.3 and 4.1 to be even more clear what speciality coffee brings above vertified coffee. E.g. Utz/Rainforest Alliance certified is well known for its farmers trainings and its collaboration with TechnoServe.
2.It is not veryy clear why farmers also do side-sales to middlemen: what is the percentage ofside-sales, do they receive better prices or are they more quickly paid (457-485)? (What does this tell about the functioning of the cooperatives?) Has this been addressed in the group interviews? (Were the group interviews only with cooperative leaders or also with non-board members?)

Validity of the findings

See above.

Additional comments

Some smaller comments:
1. What do you mean with Fair Trade (99)? Utz ceritified and Rainforest Alliance were already much larger than the Fair Trade alliance before the two (Utz and Rainforest) merged.
2. 167-168: Ruben & Hoebink (2015) present ample evidence of coffee certification also in Ethiopia as well as on the functioning of coffee cooperatives in Ethiopia.

---

## Round 0.2 · accepted · Accept

The comments made by the reviewers were satisfactorily addressed. The current version of the manuscript can be accepted for publication. Congratulations to the authors!